# Molecular characterization of *Rhipicephalus microplus* and *Haemaphysalis bispinosa* ticks from cattle across Thailand: Regional identification and evidence of different genetic sub-structures between mainland and peninsular populations

Danai Sangthong[1], Pradit Sangthong[2], Warin Rangubpit [1,3], Prapasiri Pongprayoon[3,4], Eukote Suwan[1], Kannika Wongpanit[5], Wissanuwat Chimnoi[6], Pacharathon Simking[7], Sinsamut Sae Ngow[7], Serge Morand [8], Roger W. Stich [9], Sathaporn Jittapalapong [1]*

**1** Department of Veterinary Technology, Faculty of Veterinary Technology, Kasetsart University, Bangkok, Thailand, **2** Department of Genetics, Faculty of Science, Kasetsart University, Bangkok, Thailand, **3** Department of Chemistry, Faculty of Science, Kasetsart University, Bangkok, Thailand, **4** Center for Advanced Studies in Nanotechnology for Chemical, Food and Agricultural Industries, KU Institute for Advanced Studies, Kasetsart University, Bangkok, Thailand, **5** Department of Agriculture and Resources, Faculty of Natural Resources and Agro-Industry, Chalermphrakiat Sakon Nakhon Province Campus, Kasetsart University, Sakon Nakhon, Thailand, **6** Department of Parasitology, Faculty of Veterinary Medicine, Kasetsart University, Bangkok, Thailand, **7** Faculty of Agricultural Technology, Rajamangala University of Technology Thanyaburi, Pathum Thani, Thailand, **8** IRL HealthDEEP, CNRS – Kasetsart University – Mahidol University, Faculty of Veterinary Technology, Kasetsart University, Bangkok, Thailand, **9** Knipling-Bushland United States of America Livestock Insects Research Laboratory Cattle Fever Tick Research Unit Agricultural Research Service N. Moorefield Rd., Edinburg, Texas, United States of America

* fvetspj@ku.ac.th

## Abstract

Phylogenetic and population genetic analyses were conducted on tick specimens collected from cattle in northern, northeastern, central, and southern regions of Thailand. Morphological identification indicated these ticks consisted of three species, *Rhipicephalus microplus* from all four regions, *R. sanguineus* from the northern and northeastern regions, and a *Haemaphysalis* species only collected from the northeastern region. Analysis of cytochrome c oxidase subunit I gene (*COI*) sequences identified *R. microplus* clades A and C, while clade B was not detected in this study. The same analysis indicated specimens morphologically identified as *Haemaphysalis* were *H. bispinosa,* confirming previous reports of their prevalence in northeastern Thailand. *H. bispinosa* showed low haplotype and nucleotide diversity, suggesting either a bottleneck or founder effect. Both *R. microplus* clades displayed high haplotype diversity and low nucleotide diversity, a pattern associated with population expansion. Genetic structural analysis revealed significant genetic differences in *R. microplus* clade A, especially between mainland (northern, northeastern, and central regions) and peninsular (southern region) populations, which indicated limited

**Data availability statement:** All relevant data are within the manuscript and its Supporting Information files.

**Funding:** Professor Sathaporn Jittapalapong, the head of the project and as the corresponding author of this manuscript was only one who received this funding. The funding was from the Kasetsart University Research and Development Institute (KURDI) under Grant # FF(KU)14.65 and FF(S-KU)6.66 and Kasetsart University Reinventing University Program 2021 for post-doctoral. "The funders had no role in study design, data collection and analysis, decision to publish, or preparation of the manuscript".

**Competing interests:** The authors have declared that no competing interests exist.

gene flow between these areas while suggesting movement of these ticks across the mainland. The sequence analyses described in this report enhance understanding of the natural history of ticks in Thailand and are expected to guide and strengthen tick control strategies across Southeast Asia.

## Introduction

Ticks cause substantial economic losses to limited-resource farming communities, especially in tropical and subtropical regions where approximately 80% of the world's cattle are raised [1,2]. Ixodid tick infestations cause direct damage due to feeding lesions and blood loss, and ticks transmit pathogens, including etiologic agents of four major arthropod-borne diseases of cattle, viz. anaplasmosis, cowdriosis, babesiosis, and theileriosis, which pose serious threats to both animal health and livestock productivity [3]. Thus, understanding local tick vector populations is fundamental to evaluation of tick-borne disease risks.

*Rhipicephalus (Boophilus) microplus*, commonly known as the "tropical cattle fever tick," is arguably the most devastating hematophagous ectoparasite of cattle and buffaloes worldwide, in large part because it vectors agents of bovine babesiosis and anaplasmosis, which are considered the two most important vector-borne diseases of cattle worldwide [4]. *Rhipicephalus microplus* has been reported in Asia, including Southeast Asian countries such as Thailand [5–8]. Substantial losses in milk production are associated with tick infestation of dairy cows in Thailand because each engorged female tick is thought to be responsible for the loss of 8.9 ml of milk and 1 g of live weight gain [9]. Moreover, tick-borne diseases have been reported among cattle and buffaloes throughout Thailand [8,10–12]. Thus, tick distributions in Thailand are expected to be indicators of tick-borne disease outbreaks and their subsequent economic impact on livestock production. However, morphologic taxonomy of *R. microplus* is challenging, particularly due to the difficulty in morphological differentiation at the subspecies level [13]. Conversely, molecular characterization enables distinguishing closely related, morphologically identical taxa [14].

Acarine mitochondrial *cytochrome c oxidase subunit I* gene (*COI*) sequences from different parts of the world have become increasingly available in GenBank, allowing more robust phylogenetic comparisons [15]. To date, three distinct *COI* lineage assemblages are reported within the species *R. microplus*. Additionally, *COI* was identified as a suitable genetic marker for tick species identification, with five phylogenetic clades within a putative *R. microplus* complex: *R. annulatus*, *R. australis*, and three *R. microplus* s.l. clades (A, B and C) [16]. *COI* sequence analysis was useful in the classification and identification of these ticks [17,18]. However, genetic information on cattle ticks in Thailand remains limited and is primarily based on *COI* and *16S rDNA* sequences from the northeastern region of the country. Analysis of *COI* sequences in this area revealed *R. microplus* clades A and C. The results obtained with *COI* and the highly conserved *16S rDNA* sequences underscored the advantages of *COI* in resolving evolutionary relationships within *R. microplus* [19].

To obtain more comprehensive genetic information, this study employed *COI* sequences to address three objectives: [1] sampling and *COI* sequence analysis for identification of cattle tick species from all regions across Thailand, [2] conducting molecular analyses in greater depth at the population level, and [3] estimating the demographic history of tick populations in Thailand. Possible causes of these genetic characteristics were identified and future trends predicted for these populations. These findings are expected to be useful for guiding tick and tick-borne disease control strategies across Southeast Asia.

## Materials and methods

### Samples

Tick specimens were collected from cattle in 32 provinces across the four geographic regions of Thailand. Ticks were thoroughly rinsed with double-distilled water (ddH$_2$O), air-dried, and preserved in 70% ethanol at room temperature until further processing. Morphological identification was performed to differentiate the tick species, using previous publications as references [13,20–25]. All animal care and experimental procedures were approved by the Animal Experiment Committee of Kasetsart University, Thailand (Approval No. ACKU64-VTN-018), and were conducted in strict accordance with the Regulations on Animal Experiments at Kasetsart University. This field study involving vertebrate animals fully complied with these institutional regulations.

### DNA extraction and PCR amplification

Ticks preserved in 70% ethanol were individually washed in ddH$_2$O on a sterile plate before genomic DNA was isolated with the DNeasy Blood and Tissue Kit (Qiagen, Germany), and stored at −20 °C. Primers RmicCoI_parH1 (CTCAACTA-ATCATAAAGACATTGG) and RmicCoI_parL1 (TATAAACTTCAGGGTGGCCAA) were used to amplify the *COI* target sequence from each specimen. The optimized PCR consisted of 20 µL reactions containing 2 µL of 10X Taq Buffer, 1.25 mM MgCl$_2$, 0.2 mM of each dNTP, 2 units Taq DNA polymerase (Thermo Fisher Scientific, Lithuania, EU), 0.250 µM of each primer, and 1 µL of each DNA template. The thermal cycler parameters included initial denaturation at 94 °C for 2 min, followed by 35 cycles of denaturation at 94°C for 30 s, annealing at 55 °C for 30 s, elongation at 72 °C for 30 s, and final elongation at 72 °C for 30 s. All PCR products were electrophoresed in 1% agarose gels. Amplicons were purified using the FavorPrep™ Gel/PCR Purification Kit (Favorgen, Taiwan) and submitted for Sanger DNA sequencing (ATGC Co.,Ltd., Thailand).

### Phylogenetic analysis

*COI* sequences amplified from *Rhipicephalus microplus* (94 amplicons) and *Haemaphysalis* sp. (12 amplicons) were compared to 42 tick *COI* sequences retrieved from GenBank (S1 Table). These sequences were aligned, with *Ixodes ricinus* as the outgroup, using Muscle software [26]. A neighbor-joining (NJ) tree was reconstructed with $10^5$ bootstrap replications using the program PAUP v.4 (build 169) [27].

### Population genetic analysis

Population genetics were based on the northern, northeastern, central, and southern geographic regions of Thailand. To determine the genetic structure of tick populations across Thailand, the number of haplotypes (*h*), haplotype diversity (*Hd*), nucleotide diversity (*Pi*), analysis of molecular variance (AMOVA), pairwise fixation index ($F_{ST}$), and gene flow (Nm) were analyzed using Arlequin v3.5.1.2 [28]. Genetic relationships between *COI* haplotypes of ticks in Thailand were accessed by reconstruction of Median Joining (MJ) Network using the program Network v10.2 (https://www.fluxus-engineering.com). Neutrality indices, including Tajima's D, Fu's Fs, and the divergence time parameter (τ), were determined using Arlequin v3.5.1.2. Historical demographic fluctuations were elucidated using a Bayesian Skyline Plot (BSP) in

BEAST 2.6.0 [29]. A mutation rate of 3.54% per million years [30] was applied using the HKY substitution model, which was identified as the best-fitting model by jModelTest software [31]. An MCMC method with 100 million iterations under a relaxed exponential clock was performed. Effective sample size (ESS) and the construction of the BSP were determined using TRACER 1.6 software [32].

## Results

### Morphological survey of ticks on cattle in Thailand

A total of 341 tick specimens were manually classified, using morphological keys [13,20–25], into three species: *Rhipicephalus microplus*, *R. sanguineus*, and a member of the genus *Haemaphysalis*. Morphological keys used for species identification are summarized as follows:

- *Rhipicephalus (Boophilus) microplus*: the basis capitulum was dorsally hexagonal, laterally produced, and equal in length to the mouthparts. The second palpal segments were not laterally produced but bore ridges on both the dorsal and ventral surfaces. This subgenus lacked festoons. Males possessed an adanal shield, an accessory adanal shield, and this species possessed a caudal process.

- *Rhipicephalus sanguineus*: the basis capitulum was dorsally hexagonal, laterally produced, and equal in length to the mouthparts. The second palpal segments were not laterally produced and lacked ridges on both the dorsal and ventral surfaces. This subgenus had festoons, and males possessed an adanal shield and an accessory adanal shield.

- *Haemaphysalis*: the second segment of the palp was laterally produced beyond the margin of the basis capitulum. The basis capitulum was rectangular in shape. This genus had festoons. Males did not possess an adanal shield, accessory adanal shield, or caudal process.

The prevalence of different species collected indicated that *R. microplus* was the most common species collected from cattle (78.59%), followed by members of the genus *Haemaphysalis* (16.72%) and *R. sanguineus* (4.69%). All three species were present in the northeastern region, while *R. microplus* and *R. sanguineus* were found in the northern region and *R. microplus* alone was collected in the central and southern regions (Table 1).

### Molecular identification

Representative specimen samples were used for molecular analyses to confirm morphological identifications and to more specifically identify *Haemaphysalis* and *R. microplus* species and strains, respectively. The *COI* target sequence was not amplified from *R. sanguineus.* Details for each region, subregion and their respective provinces are presented in Table 2. Amplified *COI* sequences were compared to reference sequences and tree topology was clearly separated into two *COI* clades of *Haemaphysalis* and *R. microplus*. A monophyletic clade was formed between the samples identified morphologically as *Haemaphysalis* in this study and *H. bispinosa* (OP383037) from China, clustering separately from other *Haemaphysalis* spp. sequences (Fig 1). A monophyletic relationship was also observed between *COI* sequences of Thai ticks morphologically identified as *R. microplus* (Fig 1), which aligned most closely with Clades A and C. Clade A, which consisted of *R. microplus* sequences from all regions of Thailand, was the most common genetic lineage detected in this study. Clade B was not detected among sequences tested in this study. Clade C was only detected in samples collected from the northern region.

### Genetic diversity

Nucleotide diversity analysis of these sequences revealed a consistent pattern of low *COI* nucleotide diversity among both *R. microplus* and *H. bispinosa* across all four regions surveyed, while *R. microplus* and *H. bispinosa* respectively showed low and high haplotype diversity (Fig 2). *H. bispinosa* from the northeastern region exhibited a haplotype diversity

**Table 1. Morphological identification of tick specimens collected from cattle in the four major geographic regions across Thailand. These specimens were morphologically identified with established taxonomic keys.**

| Sample Site | | Morphological Identification | | |
|---|---|---|---|---|
| Region | Province | *R. microplus* | *R. sanguineus* | *Haemaphysalis* sp. |
| North | Chiang Rai | 8 | | |
| | Chiang Mai | 27 | 4 | |
| | Phrae | 11 | | |
| | Nan | 50 | | |
| | Tak | 10 | | |
| | Region Total | 106 | 4 | |
| Northeast | Sakon Nakhon | 46 | | 24 |
| | Chaiyaphum | | 10 | |
| | Nakhon Phanom | | | 5 |
| | Nakhon Ratchasima | 5 | | |
| | Khon Kaen | | | 5 |
| | Roi Et | | | 4 |
| | Yasothon | | 2 | 6 |
| | Maha Sarakham | 2 | | |
| | Ubon Ratchathani | | | 13 |
| | Region Total | 53 | 12 | 57 |
| Central | Nakhon Pathom | 10 | | |
| | Ratchaburi | 15 | | |
| | Phetchaburi | 6 | | |
| | Saraburi | 10 | | |
| | Sa Kaeo | 20 | | |
| | Kanchanaburi | 2 | | |
| | Nakhon Sawan | 3 | | |
| | Region Total | 66 | | |
| South | Prachuap Khiri Khan | 10 | | |
| | Chumphon | 6 | | |
| | Surat Thani | 10 | | |
| | Songkhla | 6 | | |
| | Phatthalung | 8 | | |
| | Trang | 2 | | |
| | Satun | 1 | | |
| | Region Total | 43 | | |
| Sum totals | | 268 (78.59%) | 16 (4.69%) | 57 (16.72%) |

of 0.439 and a nucleotide diversity of 0.00105. *R. microplus* clade A across Thailand showed haplotype diversity ranging from 0.670 to 0.956 and nucleotide diversity ranging from 0.00151 to 0.00531. *R. microplus* clade C exhibited a haplotype diversity of 0.818 and a nucleotide diversity of 0.00105. Haplotype analysis of 182 sequences revealed three haplotypes for *H. bispinosa*, 51 for *R. microplus* Clade A, and seven for *R. microplus* Clade C. Among the 51 haplotypes of *R. microplus* Clade A, Hap_A03 was most common, with a frequency of 37.74%, and was found in all except the southern region of Thailand. The other 50 haplotypes consisted of 15 shared haplotypes, all present in adjacent regions, and the remaining 35 were region-specific haplotypes (S2 Table).

**Table 2. Sampling locations of tick specimens used for *COI* sequence analysis. Numbers represent specimens of *R. microplus* and *H. bispinosa* collected throughout Thailand. Sources include original data from this study and previously published data [33].**

| Region | Subregion | Province | R. microplus | | H. bispinosa | Source |
|---|---|---|---|---|---|---|
| | | | Clade A | Clade C | | |
| North | Upper | Chiang Mai | 5 | 6 | | This study |
| | | Chiang Rai | 4 | | | This study |
| | | Nan | 10 | 5 | | This study |
| | | Total | 19 | 11 | | |
| | Lower | Nakhon Sawan | 1 | | | This study |
| | | Tak | 7 | | | This study |
| | | Total | 8 | | | |
| | Total | | 27 | 11 | | |
| Northeast | Upper | Buengkan | 6 | | | Thinnabut, *et al*. [33] |
| | | Loei | 15 | | | Thinnabut, *et al*. [33] |
| | | Mukdahan | 10 | | | Thinnabut, *et al*. [33] |
| | | Nakhon Phanom | | | 2 | This study |
| | | Nakhon Phanom | 5 | | | Thinnabut, *et al*. [33] |
| | | Nong Bua Lamphu | 2 | | | Thinnabut, *et al*. [33] |
| | | Nong Khai | 7 | | | Thinnabut, *et al*. [33] |
| | | Sakon Nakhon | 18 | | 8 | This study |
| | | Sakon Nakhon | 5 | | | Thinnabut, *et al*. [33] |
| | | Udonthani | 2 | | | Thinnabut, *et al*. [33] |
| | | Total | 70 | | 10 | |
| | Lower | Maha Sarakham | 2 | | | This study |
| | | Maha Sarakham | 2 | | | Thinnabut, *et al*. [33] |
| | | Kalasin | 5 | | | Thinnabut, *et al*. [33] |
| | | Khon Kaen | 6 | | | Thinnabut, *et al*. [33] |
| | | Roi-et | 11 | | | Thinnabut, *et al*. [33] |
| | | Yasothon | | | 1 | This study |
| | | Ubon Ratchathani | | | 1 | This study |
| | | Nakhon Ratchasima | 3 | | | |
| | | Total | 29 | | 2 | |
| | Total | | 99 | | | |
| Central | | Sa Kaeo | 5 | | | This study |
| | | Saraburi | 2 | | | This study |
| | | Prachuap Khiri Khan | 1 | | | This study |
| | | Ratchaburi | 5 | | | This study |
| | | Phetchaburi | 6 | | | This study |
| | Total | | 19 | | | |
| South | Upper | Chumphon | 1 | | | This study |
| | | Surat Thani | 7 | | | This study |
| | | Total | 8 | | | |
| | Lower | Phatthalung | 2 | | | This study |
| | | Satun | 1 | | | This study |
| | | Songkhla | 2 | | | This study |
| | | Trang | 1 | | | This study |
| | | Total | 6 | | | |
| | Total | | 14 | | | |

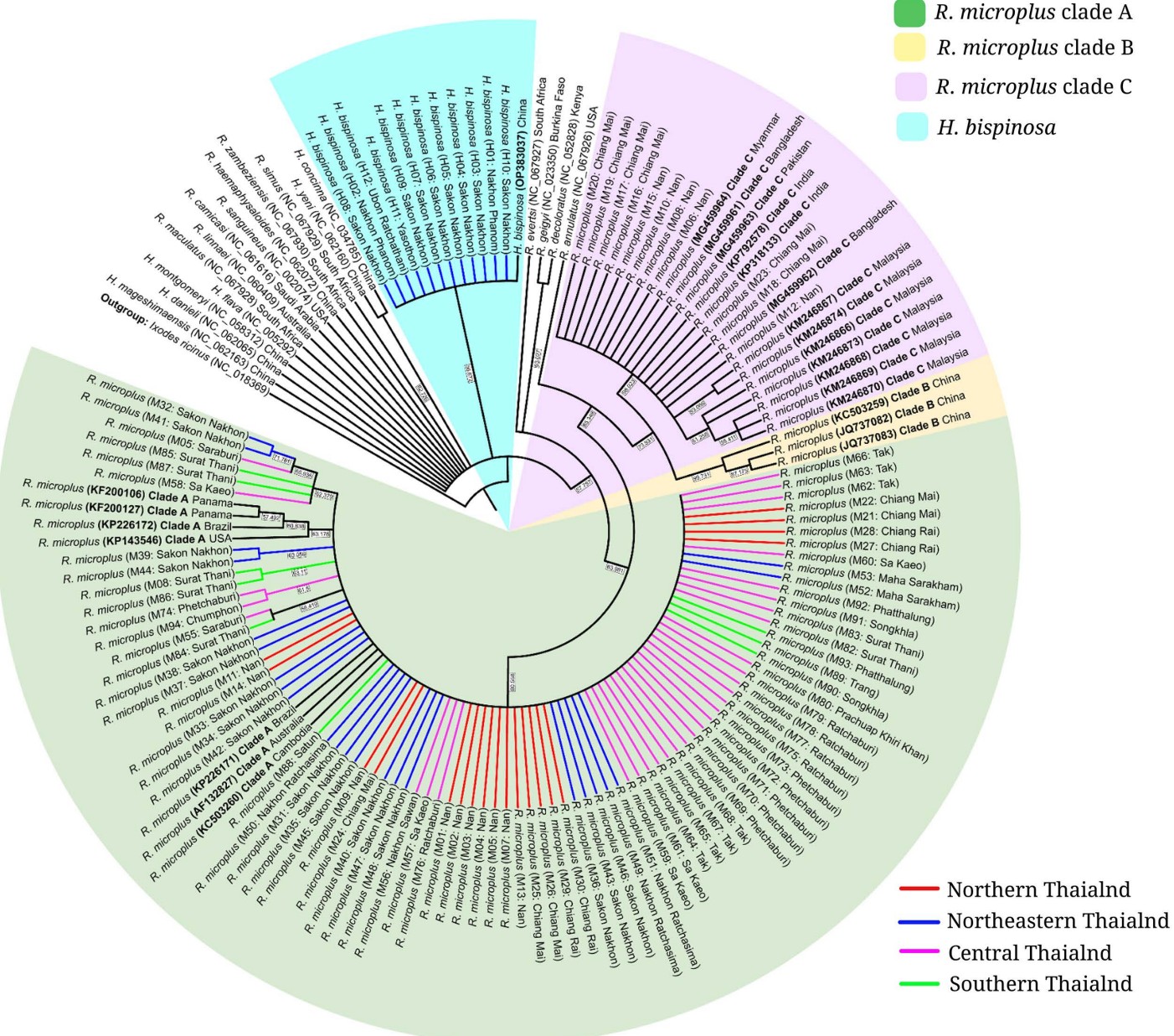

**Fig 1. Radial phylogenetic tree of tick *COI* sequences.** The tree was constructed using PAUP* software under the General Time Reversible (GTR) model, incorporating a Gamma-distributed rate of evolution and a proportion of invariant sites (GTR+G+I). Rate heterogeneity among sites was modeled using a Gamma distribution (shape parameter=0.6550), with 49.60% of sites designated as invariant (pinvar=0.4960). The tree depicts evolutionary relationships among tick species collected from various regions of Thailand. Bootstrap support values (based on 100,000 replicates) are shown at the nodes. Phylogenetic clades identified in this study are labeled as *Rhipicephalus microplus* clades A, B, and C, and *Haemaphysalis bispinosa*, represented by green, yellow, violet, and sky blue, respectively. Geographic origins of the tick samples—northern, northeastern, central, and southern Thailand—are indicated by branch colors: red, blue, pink, and green, respectively.

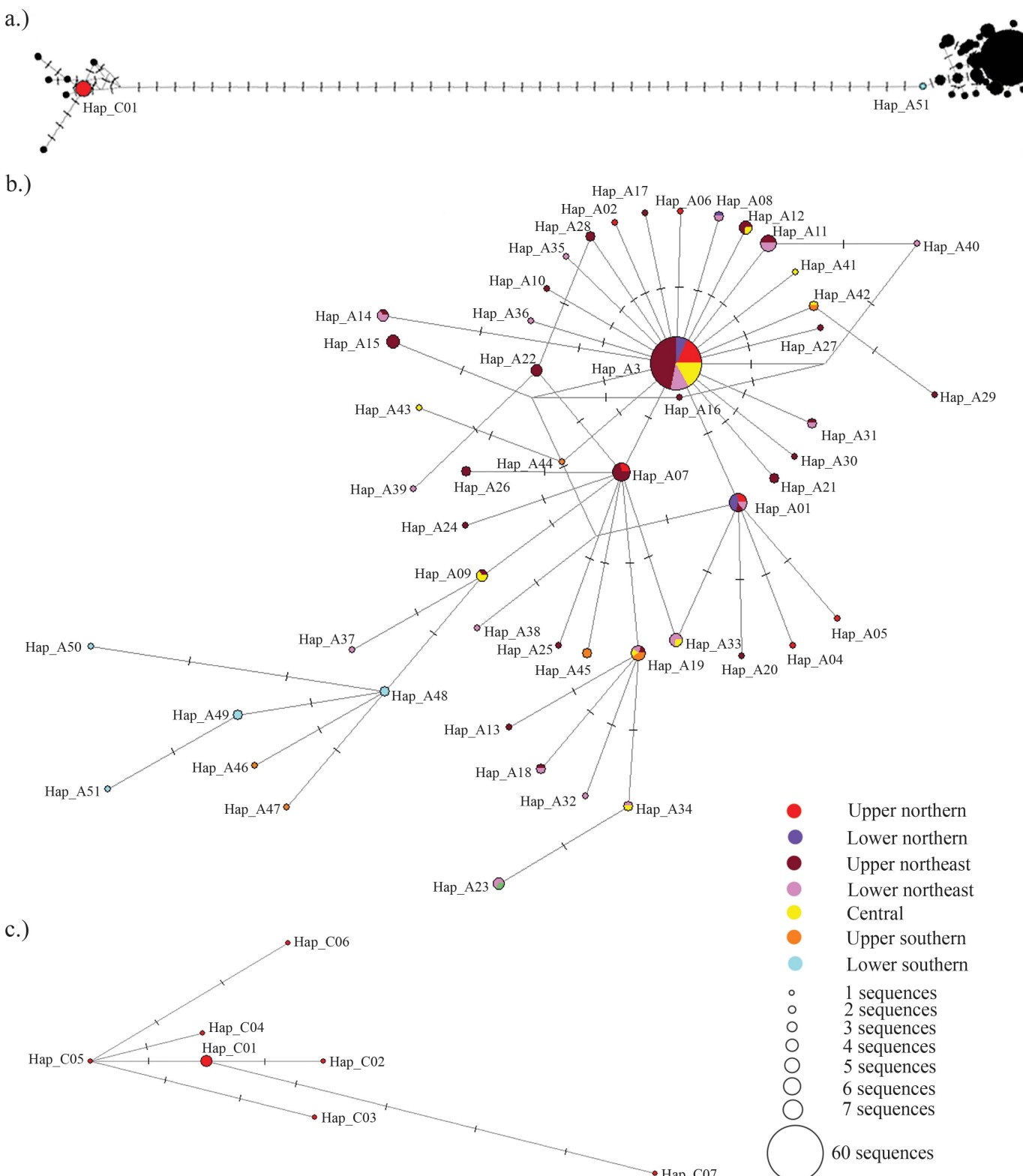

**Fig 2. Median joining network of *R. microplus* haplotypes from Thailand based on *COI* sequences, constructed using Network v10.2 software.** Circle sizes represent the number of sequences belonging to each haplotype. Vertical bars on lines connecting haplotypes indicate the number of

substitutions separating them. The geographic origins of each haplotype are indicated by different colors. The network relationship between *R. microplus* clades A and C is shown in **(a)**. Connection details between haplotypes within clades A and C are presented in (b) and **(c)**, respectively.

## Genetic structure

Genetic differences between tick populations from different geographic regions in Thailand were analyzed based on various hypotheses of population structure (Table 3). The AMOVA results indicated genetic structure only when applying the two-region hypothesis, which divided the regions into the mainland (northern, northeastern and central) and peninsula (southern). Genetic variation between the mainland and peninsula accounted for 33.23% of the total variation and showed significant F-statistics ($F_{CT}$=0.332, $p$=0.048). Genetic differences between the mainland and peninsular populations were reassessed using pairwise $F_{ST}$ comparisons (Table 4), which ranged from 0.162 to 0.740, all of which were significant ($p \leq 0.05$). In contrast, the pairwise $F_{ST}$ values between populations within the mainland (northern, northeastern, and central) were non-significant, ranging from −0.019 to 0.076. However, the $F_{ST}$ comparison between the mainland and peninsula showed a significant difference ($F_{ST}$=0.350, $p \leq 0.05$). The Nm values among the seven geographic regions ranged from 0.176 to infinity (Table 4). Nm values between the Lower Southern population and all other populations were below 1.00, while those for the Upper Southern population were higher, ranging from 1.227 to 5.212. In contrast, Nm values among the five mainland populations ranged from 6.091 to infinity.

## Demographic history

Neutrality tests based on Tajima's D and Fu's Fs were conducted (Table 5). Negative values of Tajima's D and Fu's Fs were observed for *H. bispinosa*, which were collected only from northern Thailand, and from all regions for *R. microplus* clades A and C. Significant negative values of Tajima's D test were detected in *R. microplus* clade A from the northeast and central regions, while significant and highly significant values of Fu's Fs were detected in all populations of *R. microplus*. Pooled population analysis of *R. microplus* clade A, from all regions of Thailand, showed highly significant Tajima's D and Fu's Fs values of −2.20001 and −27.29267, respectively. A significantly negative value of Tajima's D test indicated purifying selection or population expansion, while a significantly negative value of Fu's Fs directly indicated population expansion. To assess population expansion, *R. microplus* sequences representing both clades were further analyzed using a skyline plot, with a substitution rate of 0.0354 per site per million years. BSP analysis revealed that the effective population size of *R. microplus* was stable over long periods of time and initiated a sudden expansion in the last 25,000 years (Fig 3).

## Discussion

Thailand is an agricultural country with a diverse range of sectors. The livestock sector is a major industry with high market value, providing benefits to all levels of society [34,35]. However, tropical livestock production faces challenges, one of which is tick infestations [36]. At least 25 ixodid tick species have been identified in Thailand, belonging to the genera *Haemaphysalis*, *Ixodes*, *Amblyomma*, *Dermacentor*, *Rhipicephalus* and *Nosomma* [37]. In terms of threats to livestock health, particularly for cattle, only a few tick species have been reported, including *R. haemaphysaloides*, *R. microplus*, *R. sanguineus* and *H. bispinosa* [33,38,39].

Morphological identification results in this study revealed that *R. microplus* was prevalent throughout Thailand, suggesting *R. microplus* is a source of significant economic losses in livestock production due to its widespread distribution and ability to transmit pathogens such as *Babesia bigemina*, *B. bovis* and *Anaplasma marginale* [40]. Notably, *R. microplus*, *R. sanguineus*, and *H. bispinosa*, which are considered common tick species among livestock in Thailand, were identified among the specimens collected in this study. In contrast, *R. haemaphysaloides* was not detected in the current study, although a total of nine specimens of this species were previously reported from four of 18 provinces in the northeast region

**Table 3. Hierarchical analysis of molecular variance (AMOVA) of 4 posited genetic structures. Bold values are statistically significant.**

| Source of variation | d.f. | Sum of Squares | Variance components | Percentage of variation | p value |
|---|---|---|---|---|---|
| 1) four regions: North (upN, loN), Northeast (upNE, loNE), Central and South (upS, loS) | | | | | |
| Among regions | 3 | 17.171 | 0.106 | 9.58 | $F_{CT}=0.96$ (p=0.132) |
| Among populations within regions | 3 | 7.743 | 0.084 | 7.58 | $F_{SC}=0.084$ (p=0.002) |
| Within populations | 152 | 139.992 | 0.921 | 82.85 | $F_{ST}=0.172$ (p=0.000) |
| Total | 158 | 164.906 | 1.112 | | |
| 2) three regions: North (upN, loN, upNE), Middle (loNE, C) and South (upS, loS) | | | | | |
| Among regions | 2 | 15.265 | 0.115 | 10.26 | $F_{CT}=0.103$ (p=0.067) |
| Among populations within regions | 4 | 9.651 | 0.083 | 7.42 | $F_{SC}=0.083$ (p=0.001) |
| Within populations | 152 | 139.992 | 0.921 | 82.32 | $F_{ST}=0.177$ (p=0.000) |
| Total | 158 | 164.906 | 1.119 | | |
| 3) two regions: Upper (upN, loN, upNE, loNE) and Lower (C, upS, loS) | | | | | |
| Among regions | 1 | 6.647 | 0.052 | 4.64 | $F_{CT}=0.046$ (p=0.058) |
| Among populations | 5 | 18.267 | 0.142 | 12.75 | $F_{SC}=0.134$ (p=0.000) |
| Within populations | 152 | 139.992 | 0.921 | 82.60 | $F_{ST}=0.174$ (p=0.000) |
| Total | 158 | 164.906 | 1.115 | | |
| 4) two regions: Mainland (upN, loN, upNE, loNE, C) and Peninsula (upS, loS) | | | | | |
| Among regions | 1 | 13.993 | 0.488 | 33.23 | ***FCT=0.332 (p=0.048)*** |
| Among populations | 5 | 10.920 | 0.059 | 4.03 | $F_{SC}=0.060$ (p=0.001) |
| Within populations | 152 | 139.992 | 0.921 | 62.75 | $F_{ST}=0.373$ (p=0.000) |
| Total | 158 | 164.906 | 1.468 | | |

**Table 4. Pairwise $F_{ST}$ values (lower diagonal) and gene flow Nm (upper diagonal) between tick populations from different regions of Thailand. Values in the lower diagonal represent genetic differentiation ($F_{ST}$), while those in the upper diagonal indicate estimated gene flow (Nm) between populations. "Inf" denotes infinite gene flow. Statistically significant $F_{ST}$ values are shown in bold.**

| | Upper north | Lower north | Upper northeast | Lower northeast | Central | Upper south | Lower south |
|---|---|---|---|---|---|---|---|
| Upper north | | inf | 19.726 | 15.456 | 18.739 | 1.227 | 0.185 |
| Lower north | −0.019 | | 7.689 | 17.223 | 6.091 | 1.250 | 0.176 |
| Upper northeast | 0.025 | **0.061** | | 42.620 | inf | 2.592 | 0.334 |
| Lower northeast | 0.031 | 0.028 | 0.012 | | inf | 5.212 | 0.504 |
| Central | 0.026 | 0.076 | −0.010 | −0.003 | | 3.696 | 0.309 |
| Upper south | **0.290** | **0.286** | **0.162** | **0.088** | **0.119** | | 0.930 |
| Lower south | **0.730** | **0.740** | **0.600** | **0.498** | **0.618** | **0.350** | |

**Table 5. Summary of *COI* diversity of *H. bispinosa* and *R. microplus* clades A and C.**

| | N[A] | h[B] | Hd[C] | Pi[D] | tau[E] | Tajima's D | Fu's Fs |
|---|---|---|---|---|---|---|---|
| *H. bispinosa* | | | | | | | |
| Northeast | 12 | 3 | 0.439 | 0.00105 | 0.813 | −1.17901 | −0.18049 |
| *R. microplus* clade A | | | | | | | |
| North | 27 | 8 | 0.670 | 0.00151 | 1.045 | −1.50990 | −4.63189** |
| Northeast | 99 | 36 | 0.887 | 0.00342 | 2.195 | −2.11646** | −27.31337** |
| Central | 19 | 9 | 0.731 | 0.00256 | 2.174 | −1.63282* | −4.66710** |
| South | 14 | 10 | 0.956 | 0.00531 | 3.609 | −0.86414 | −4.28379** |
| Totals | 159 | 51 | 0.850 | 0.00346 | 2.180 | −2.20001** | −27.29267** |
| *R. microplus* clade C | | | | | | | |
| Northern | 11 | 7 | 0.818 | 0.00385 | 3.17578 | −1.63649* | −2.40320* |

[A]Number of specimens.

[B]Number of haplotypes.

[C]Haplotype diversity.

[D]Nucleotide diversity.

[E]Divergence time parameter.

of Thailand [19]. Collectively, these reports support previous suggestions that *R. haemaphysaloides* was rarely found, only in the northeast region of Thailand [19], and suggests its limited presence in other regions. This distribution pattern aligns with reports of *R. haemaphysaloides* prevalence in Bhutan [41], China [42,43], and Sri Lanka [44]. Despite its limited distribution, *R. haemaphysaloides* remains a potential vector of disease agents and warrants continued surveillance [45].

Mitochondrial *COI* sequences were used to expand upon morphological identification of two species through molecular analysis, *H. bispinosa* and *R. microplus*. Other groups have used these sequences for specific identification of morphologically identical specimens, including a *COI* lineage restricted to *R. microplus* from China, which those authors suggested could be a cryptic species more closely related to *R. annulatus* [16]. In the current study, phylogenetic tree topology revealed a monophyletic clade of 12 *Haemaphysalis* sequences, which aligned most closely with the *H. bispinosa* reference sequence (OP383037). Taxonomic classification beyond the genus *Haemaphysalis* in Thailand, based on morphological characteristics, has been challenging. As a result, several authors reported such samples as "*Haemaphysalis* sp." [46] or "*Haemaphysalis* spp." [47–49]. The *COI* analysis performed in this study, incorporating a novel pair of primers and PCR protocol, overcame the limitations of morphological identification of *Haemaphysalis* spp., indicating these 12 *Haemaphysalis* sequences were *H. bispinosa* and the establishment of this species in northeastern Thailand.

The phylogenetic tree demonstrated *COI*-based identification of *R. microplus* aligned with morphological findings while offering enhanced resolution to the subspecies level. All 170 *R. microplus* sequences analyzed in this study were distinctly grouped into two clades, 159 sequences in clade A and 11 in clade C. *R. microplus* clade A was the most common across the four geographic regions sampled in Thailand, corroborating previous reports of its high prevalence in specific surveillance areas, such as the northeast region [19]. *R. microplus* clade A has been reported as the dominant and most widely distributed clade across subtropical and tropical regions worldwide [6,50–53]. Clade C, the other *R. microplus* lineage detected in this study, was also less frequent than clade A in other areas, including Pakistan, Myanmar, Malaysia, Bangladesh, India and Thailand [54–56]. Previously, 91 and 25 specimens of *R. microplus* clades A and C, respectively, were reported in studies focused on the northeastern region of Thailand [19]. However, another study in the same region detected only *R. microplus* clade A among the 79 *R. microplus* specimens examined [33]. In the present study, *R. microplus* clade C was found only in the northern region. The distribution of *R. microplus* clade C in Thailand, as well as in sub-regions of other countries where clade C was identified, warrants further investigation.

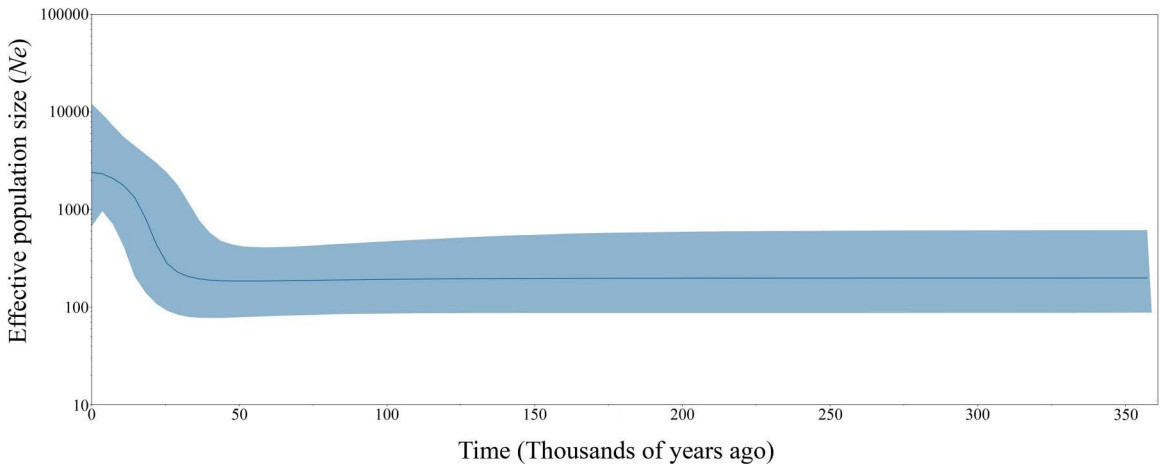

**Fig 3. Demographic history of *Rhipicephalus microplus* in Thailand inferred using a Bayesian skyline plot based on *COI* sequences.** The x-axis represents time in thousands of years before present (x 1000 years ago) and the y-axis represents the effective population size on a logarithmic scale. The blue solid line indicates the mean estimate, while the copper-blue shaded area represents the 95% highest probability density (HPD) interval.

*R. microplus* clade B, which some authorities consider a cryptic species indigenous to Southern China and Northern India, was not detected in the current study and has not been reported in Thailand [19,33]. Based on this detailed investigation into the genetic lineages of *R. microplus* in Thailand, clade A is most likely the primary contributor to tick-borne disease outbreaks and the associated economic losses in this country. The less frequent clade C has an uncertain prevalence in Thailand, and its impact on livestock remains unclear. However, previously reported occurrences of clades B and C suggested they could potentially pose a threat in Thailand. Clade B has been reported in several states in northern India (Rajasthan, Punjab, Uttar Pradesh, Bihar, and West Bengal states) and southern China, while clade C has been widely reported in Pakistan, India, Bangladesh, Myanmar, and Malaysia [56]. These areas have continuous live cattle trade with Thailand, which likely facilitates the spread of ticks from all genetic lineages. Therefore, although clade B is currently undetectable and clade C remains less common, they could become more important due to the trade flows of live cattle across South and Southeast Asia, because Thailand imports live cattle from the west (India, Bangladesh and Myanmar) and exports them to several regions, including to the north (China and Laos), east (Cambodia and Vietnam) and south (Malaysia) [57]. Thus, trade routes could facilitate the spread of clades B and C that originate in other countries along Thailand's western trade routes.

Haplotype analysis revealed clear distinctions between *H. bispinosa* and *R. microplus*, as well as between *R. microplus* lineages A and C, without haplotypes shared between clades or species. The absence of shared haplotypes among ticks in this study indicated a lack of gene flow, suggesting that they belong to distinct reproductive groups [58,59]. Although the lack of shared haplotypes may be attributed to the small sample size in some analysis groups [60], the distinct genetic groups—between *H. bispinosa* and *R. microplus*, as well as between *R. microplus* clades A and C—remain supported by the results of phylogenetic inference and network analysis. The haplotype diversity and nucleotide diversity of *H. bispinosa* and *R. microplus* clades A and C, based on the samples collected in this study, represent different categories of genetic diversity. *H. bispinosa* exhibited low values for both haplotype diversity ($Hd < 0.5$) and nucleotide diversity ($Pi < 0.5$), suggesting this tick species recently experienced a population bottleneck or a founder effect [61]. Conversely, *R. microplus* clades A and C showed high values of haplotype diversity ($Hd > 0.5$) and low values of nucleotide diversity ($Pi < 0.5$), presumably due to recent expansion of this tick population [62].

Although population expansion of ticks is commonly associated with recent cattle trade flows that facilitate the spread of ticks through live cattle transfers, the skyline plot in the present study suggested an expansion began over 25,000 years ago, predating modern trade routes. The initiation of the expansion period for *R. microplus* was estimated to have occurred during the late Pleistocene epoch, approximately 129,000–11,700 years ago [63]. During this period, the region that is now Thailand served as a major route for human migration and the dispersal of various mammalian species [64–66]. The high population density and migratory activities of both hominin and non-hominin fauna, including Artiodactyla, were likely significant factors in driving the population growth of *R. microplus* during this time period.

*R. microplus* exhibited relatively high haplotype ($Hd > 0.5$) and low nucleotide ($Pi < 0.5$) diversities in both regional and overall analyses across Thailand, suggesting the *R. microplus* population in Thailand is undergoing expansion. A similar pattern of genetic diversity has been observed in tick populations globally, including Malaysia [55], Brazil [50] and the Neotropical region covering Panama, Colombia, Brazil, and Argentina [18]. These observed genetic characteristics indicated these tick populations are expanding globally as well as in Thailand.

The population structure of *R. microplus* clade A revealed distinct differences between the peninsular (southern) and mainland populations. The lack of genetic structure and high gene flow observed among mainland populations (northern, northeastern, and central regions) in this study is likely influenced by the intensive production and movement of cattle and buffalo across these areas. Additionally, connections to adjacent countries likely contributed indirectly to increased gene flow between tick populations [67]. Limited gene flow between the southern and other regions could be due to less economic activity and thus less movement of cattle and buffalo connected to the southern region [68–70]. Because genetic differences were detected between mainland and peninsular populations, further study of *R. microplus* indigenous to southern Thailand is warranted.

In summary, this report confirmed the utility of integrating morphological and molecular approaches for identification of tick species indigenous to Thailand. The use of mitochondrial *COI* sequencing not only rectified challenges of classifying morphologically similar *Haemaphysalis* spp. but also provided finer resolution at the subspecies level of *R. microplus*. These findings confirmed that *R. microplus* is the predominant species affecting cattle in Thailand, with clade A widespread among multiple regions and clade C limited to the north, while *H. bispinosa* was established in the northeast. Haplotype analyses revealed distinct genetic groups with high haplotype but low nucleotide diversities within *R. microplus* clades, indicating an expansion tracing back to the late Pleistocene epoch, a period marked by significant faunal migrations. Collectively, these results underscore the economic impact of tick infestations on Thailand's livestock sector and highlight potential risks associated with live cattle trade, which may influence the distribution of *R. microplus* clades. Further research, particularly into lineages and regional genetic structures, is warranted to develop effective management strategies for mitigation of tick-borne diseases of livestock in Thailand.

## Supporting information

**S1 Table. Reference sequences of *R. microplus*, *Rhipicephalus* spp., *Haemaphysalis* spp. and outgroup (*Ixodes ricinus*).**
(DOCX)

**S2 Table. Haplotype distribution of *H. bispinosa* and *R. microplus* Clades A and C in Thailand.**
(DOCX)

## Author contributions

**Conceptualization:** Danai Sangthong, Pradit Sangthong, Sathaporn Jittapalapong.

**Data curation:** Warin Rangubpit.

**Formal analysis:** Warin Rangubpit.

**Funding acquisition:** Sathaporn Jittapalapong.

**Investigation:** Danai Sangthong, Pradit Sangthong, Eukote Suwan, Kannika Wongpanit, Wissanuwat Chimnoi, Pacharathon Simking, Sinsamut Sae Ngow.

**Methodology:** Danai Sangthong, Pradit Sangthong, Prapasiri Pongprayoon, Eukote Suwan, Kannika Wongpanit, Wissanuwat Chimnoi, Pacharathon Simking, Sinsamut Sae Ngow.

**Project administration:** Sathaporn Jittapalapong.

**Resources:** Danai Sangthong, Pradit Sangthong.

**Software:** Warin Rangubpit, Prapasiri Pongprayoon.

**Supervision:** Serge Morand, Sathaporn Jittapalapong.

**Validation:** Roger W Stich, Sathaporn Jittapalapong.

**Visualization:** Prapasiri Pongprayoon, Roger W Stich.

**Writing – original draft:** Danai Sangthong.

**Writing – review & editing:** Serge Morand, Roger W Stich, Sathaporn Jittapalapong.

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
