## [Decision Letter · Decision Letter 0]

10 Jul 2025

Dear Dr. JITTAPALAPONG,

Thank you for submitting your manuscript to PLOS ONE. After careful consideration, we feel that it has merit but does not fully meet PLOS ONE’s publication criteria as it currently stands. Therefore, we invite you to submit a revised version of the manuscript that addresses the points raised during the review process.

**ACADEMIC EDITOR:** The authors need to revise the manuscript in accordance to the comments of the reviewers with special attention to why the authors used one gene in the molecular identification of ticks? Also, the signficant of the study findings of the current manuscript. 

We look forward to receiving your revised manuscript.

Kind regards,

Shawky M Aboelhadid, PhD

Academic Editor

PLOS ONE

Journal Requirements:

This work was supported by the Kasetsart University Reinventing University Program 2021 for post-doctoral training. The authors also acknowledge funding from the Kasetsart University Research and Development Institute (KURDI) under Grant Nos. FF(KU)14.65 and FF(S-KU)6.66.

Professor Sathaporn Jittapalapong, the head of the project and as the corresponding author of this manuscript was only one who received this funding.

The funding was from the Kasetsart University Research and Development Institute (KURDI) under Grant # FF(KU)14.65 and FF(S-KU)6.66 and Kasetsart University Reinventing University Program 2021 for post-doctoral.

6. Please amend either the title on the online submission form (via Edit Submission) or the title in the manuscript so that they are identical.

Reviewers' comments:

Reviewer's Responses to Questions

**Comments to the Author**

1. Is the manuscript technically sound, and do the data support the conclusions?

Reviewer #1: Yes

Reviewer #2: Yes

Reviewer #3: Yes

2. Has the statistical analysis been performed appropriately and rigorously?

Reviewer #1: Yes

Reviewer #2: Yes

Reviewer #3: Yes

3. Have the authors made all data underlying the findings in their manuscript fully available?

Reviewer #1: Yes

Reviewer #2: Yes

Reviewer #3: Yes

4. Is the manuscript presented in an intelligible fashion and written in standard English?

Reviewer #1: Yes

Reviewer #2: No

Reviewer #3: Yes

Reviewer #1: This is a very interesting study that reports on the genetic diversity and population genetics of ticks in Thailand, with a relatively good sample size. I have two main comments. First, Rhipicephalus microplus clade C was first reported in Malaysia, but the reference sequences were not included in the analysis. Including and comparing these sequences would be valuable to understand how they differ from the Thai specimens, and this point should be discussed in more detail. Second, while the authors discussed gene flow, they did not specifically analyze or report the gene flow estimate (Nm), which is important for interpreting population connectivity and should be considered in the analysis.

Reviewer #2: This research focuses on the molecular characterization of ticks collected from cattle and buffaloes in Thailand using the CO1 gene sequence. My concern is that the authors used only a single mitochondrial gene (CO1) for genotyping ticks in this study, which may not be sufficient to provide a comprehensive understanding of their genetic diversity and phylogenetic relationships. It is recommended to include additional genetic markers, such as nuclear genes or other mitochondrial regions, to strengthen the findings and improve the resolution of species identification and genetic variation.

- Title: I would suggest highlighting the main findings in the title to gain broader readership (i.e., indicating R. microplus genetic sub-structure between mainland and peninsular populations)

- Page 4, line 72: correct to be “cytochrome c oxidase subunit I”

- Page 4, line 79: The authors stated that “genetic information about ticks in Thailand is limited”, however, several studies have reported on the genetic variation of cattle ticks in northeastern Thailand. Therefore, this sentence should be revised to: “genetic information about cattle ticks in Thailand is limited and primarily restricted to the northeastern region.” and relevant findings from these previous studies should also be incorporated into the Introduction.

- Page 5, line 100: Details on morphological identification and keys to separate the species should be clearly mentioned. Which references were exactly used for identification?

- Page 5, line 97: The authors stated that ticks were collected from cattle and buffaloes; however, these details are not presented in Table 1.

- Page 7, line 125: Was the primer pair used in this study designed by the authors or adopted from a previous report?

- Page 7, line 132: Please verify whether the final elongation step at 72°C for only 30 seconds is sufficient, as it may be too short.

- Page 7, line 136, How did the author select the 33 sequences from GenBank to be included in the tree analysis. Why not include the CO1 sequences of cattle ticks from northeast Thailand in previous reports? Please clarify.

- Table 3 should be moved to the supplementary materials.

- What do the bold values in Table 5 represent? Please clarify.

- FST should be FST

- There are several typos, please carefully check throughout the manuscript.

Why was R. haemaphysalis not found in cattle as reported in a previous study? Please discuss.

- This type of research has previously been reported, particularly focusing on northeastern Thailand. The authors should provide a clearer explanation and strengthen the manuscript by comparing their data with findings from previous studies or by highlighting how their study adds new insights beyond existing reports in the Discussion section.

Reviewer #3: The manuscript presents molecular characterization data of Rh. microplus and Hae. bispinosa ticks; however, my main concern, why Rh. sanguineus are given less attention in the study although its huge veterinary importance worldwide. I would say to present the full data for this specific species as well.

The authors are invited to go through the uploaded file to find out the corrections.

**Do you want your identity to be public for this peer review?** For information about this choice, including consent withdrawal, please see our Privacy Policy

Reviewer #1: No

Reviewer #2: No

Reviewer #3: No

---

## [Author Response · Author response to Decision Letter 1]

2 Sep 2025

Dear Academic Editor:

First, we want to thank the reviewers (and Editor) for their time and thoughtful comments. The authors sincerely appreciate the constructive and insightful feedback, and it is our sincere belief that these comments resulted in a significantly improved manuscript. All comments from each reviewer have been addressed either through revisions to the manuscript or by providing explanations based on our recently obtained data and the challenges encountered during this research. In particular, regarding the two main points of concern—the use of a single gene in this study and the significance of the findings—the authors have provided a thorough explanation of the rationale and associated limitations (see Answer #2). Additionally, we have emphasized the value of the information obtained from the COI sequences, which is supported by previous studies, including those involving tick specimens collected from northeastern Thailand. Finally, three clear objectives underscore the significance of this study in the last paragraph of the Introduction section of this manuscript.

The manuscript has been revised accordingly, and our detailed responses to the reviewers’ comments are provided below.

Reviewer #1:

This is a very interesting study that reports on the genetic diversity and population genetics of ticks in Thailand, with a relatively good sample size. I have two main comments. First, Rhipicephalus microplus clade C was first reported in Malaysia, but the reference sequences were not included in the analysis. Including and comparing these sequences would be valuable to understand how they differ from the Thai specimens, and this point should be discussed in more detail. Second, while the authors discussed gene flow, they did not specifically analyze or report the gene flow estimate (Nm), which is important for interpreting population connectivity and should be considered in the analysis.

Response to Reviewer #1:

It was gratifying to read your positive comments. More clarification based on your observations is provided below.

The Nm analysis tool, results, and corresponding discussion were added to the manuscript in response to Reviewer 1’s suggestion (lines 126, 223-227 and 360–363). The Nm values are now presented in Table 4 (page 14).

The remaining concern referred to Rhipicephalus microplus reported in Malaysia, which was suggested for inclusion in this study's analysis. We obtained 17 sequences from both clades A and C of R. microplus originating from Malaysia and intended to include some as reference sequences, similar to those from other neighboring countries of Thailand, such as Cambodia and Myanmar. However, multiple sequence alignment revealed non-conserved positions in the Malaysian sequences compared to those obtained in this study and other reference sequences. When compared to the NCBI reference sequence of Rhipicephalus microplus (NC_023335), the sequences obtained in this study and the reference sequences correspond to positions 1186–1799. In contrast, the Malaysian sequences OM100586, OM100587, OM100588, and OM100589 align at positions 1256–1863, KM246866, KM246867, KM246868, KM246869, KM246870, KM246871, KM246872, KM246873, KM246874, KM246875, KM246876, and KM246877, are located at positions 1383–2008 of NC_023335. Thus, inclusion of the Malaysian sequences would introduce large gaps at both the 5′ and 3′ ends of the alignment, likely reducing the accuracy of the analysis. Importantly, although Malaysian sequences were excluded from the phylogenetic analysis, reference sequences from neighboring countries and overseas sources were included, ensuring that the results obtained are robust and reliable.

Reviewer #2:

This research focuses on the molecular characterization of ticks collected from cattle and buffaloes in Thailand using the CO1 gene sequence. My concern is that the authors used only a single mitochondrial gene (CO1) for genotyping ticks in this study, which may not be sufficient to provide a comprehensive understanding of their genetic diversity and phylogenetic relationships. It is recommended to include additional genetic markers, such as nuclear genes or other mitochondrial regions, to strengthen the findings and improve the resolution of species identification and genetic variation.

Response to Reviewer #2:

Answer #2:

The authors appreciate your suggestion regarding the use of additional genetic markers for the identification of tick species and strains. However, it is our opinion that COI sequence analysis is sufficient for identification of species and subspecies in the current study. Notably, there are multiple examples of molecular identification of ticks with a single gene sequence. Much of the literature involves one or both of two mitochondrial genes, COI and 16S rDNA. COI is a standard barcode region because it is more variable than the highly conserved genes encoding 16 rRNA, allowing better resolution for identification to the species and subspecies level.

Indeed, previous studies have employed COI in combination with other genetic markers, particularly 16S rDNA, and have demonstrated that COI produced consistent results in phylogenetic analyses and offered advantages in assessing population genetic structure by providing greater resolution and more detailed insights (Atopkin et al., 2023; Wei et al., 2023). This advantage of COI gene was also reported with tick specimens from northeastern Thailand (Tantrawatpan et al., 2022). Therefore, COI remains a reliable and informative tool—particularly when combined with rigorous statistical analysis pipelines—and should not be regarded as inadequate or lacking scientific merit.

Consequently, to maximize the reliability of the data and minimize confusion arising from incomplete or inconsistent results, we ultimately chose to focus on COI sequence analysis for the collected tick specimens.

Atopkin DM. 2023. Analysis of partial mitochondrial COI and 16S rRNA geen sequences variation of Pagurus brachiomastus and Pagurus proximus (Decapoda: Paguridae) populations from the South of Russian Far East. Genet.Aquat.Org. 7: GA574.

Tantrawatpan C, Valsusuk K, Chatan W, Pilap W, Suksavate W, Andrews RH, Petney TN, Saljuntha W. 2022. Genetic diversity and phylogenetic analyses of ixodid ticks infesting cattle in northeast Thailand: the discovery of Rhipicephalus microplus clade C and the rarely detected R. haemaphysaloides. Exp.Appl.Acarol. 56: 535-548.

Wei D, Zheng S, Wang S, Yan J, Liu Z, Zhou L, Wu B, Sun X. 2023. Genetic and haplotype diversity of manila clam Ruditapes philippinarum in different regions of china based on three molecular markers. Animals 13: 2886.

- Title: I would suggest highlighting the main findings in the title to gain broader readership (i.e., indicating R. microplus genetic sub-structure between mainland and peninsular populations)

Answer:

As per this suggestion, the title was revised to highlight the main findings as follows:

“Molecular characteristics of Rhipicephalus (Boophilus) microplus and Haemaphysalis bispinosa ticks from cattle across Thailand: Identification of cattle ticks found in different regions and evidence of different genetic sub-structures among R. microplus from mainland and peninsular regions”

- Page 4, line 72: correct to be “cytochrome c oxidase subunit I”

Answer:

The manuscript was amended according to this correction, as reflected in line 73 of the revised manuscript.

- Page 4, line 79: The authors stated that “genetic information about ticks in Thailand is limited”, however, several studies have reported on the genetic variation of cattle ticks in northeastern Thailand. Therefore, this sentence should be revised to: “genetic information about cattle ticks in Thailand is limited and primarily restricted to the northeastern region.” and relevant findings from these previous studies should also be incorporated into the Introduction.

Answer:

The manuscript has been revised accordingly, as reflected in lines 80–91 of the revised manuscript.

- Page 5, line 100: Details on morphological identification and keys to separate the species should be clearly mentioned. Which references were exactly used for identification?

Answer:

The exact references used are cited in the Methods section on line 99, and the details of the morphological characteristics of the ticks collected in this study are now described on lines 143–155.

- Page 5, line 97: The authors stated that ticks were collected from cattle and buffaloes; however, these details are not presented in Table 1.

Answer:

We thank you for your observation. The sample details in this study have been clarified, and all specimens were confirmed to have been collected from cattle, as presented in Table 1. The manuscript has been revised accordingly (lines 95–96).

- Page 7, line 125: Was the primer pair used in this study designed by the authors or adopted from a previous report?

Answer:

The primers used in this study were novel and designed by the authors (lines 106-107).

- Page 7, line 132: Please verify whether the final elongation step at 72°C for only 30 seconds is sufficient, as it may be too short.

Answer:

The components and thermal cycling conditions were carefully reviewed, and the authors confirm that they are accurately described in the manuscript.

- Page 7, line 136, How did the author select the 33 sequences from GenBank to be included in the tree analysis. Why not include the CO1 sequences of cattle ticks from northeast Thailand in previous reports? Please clarify.

Answer:

The molecular analysis conducted in this study was organized into two main sections. The first section involves the use of phylogenetic inference for subspecies or species identification and is referred to as the molecular taxonomy section. In this section, sequences retrieved from GenBank were used as reference sequences for subspecies or species previously reported in the literature. Therefore, a wide range of representative subspecies and species was included to achieve comprehensive phylogenetic resolution. However, including too many reference sequences can make it difficult to present a clear and interpretable phylogenetic tree figure. While sequences from northeastern Thailand were not directly included, their taxonomic groups were represented by reference sequences retrieved from GenBank. The analysis aimed to incorporate as many subspecies and species reference sequences as available in the GenBank database.

The second section is dedicated to the analysis of population genetics of tick populations collected in Thailand. This section includes all available sequences from Thailand in the analysis (Table 2), encompassing both newly sequenced data from this study and cattle tick sequences from northeastern Thailand retrieved from the GenBank database.

- Table 3 should be moved to the supplementary materials.

Answer:

Table 3 in the previous version has been reassigned as Table S2.

- What do the bold values in Table 5 represent? Please clarify.

Answer:

The bold values represent statistically significant FST values. The legend of Table 5 has been updated to include this information, in accordance with the reviewer’s suggestion.

- FST should be FST

Answer:

Various formatting styles for FST are used in recent publications. Therefore, the authors have chosen to use the style presented by Wright (1949), in which FST is written with all letters italicized, with 'F' in uppercase and 'ST' as uppercase subscripts.

Wright S. 1949. The genetical structure of populations. Ann Hum Genet 15: 323–354.

- There are several typos, please carefully check throughout the manuscript.

Answer:

The manuscript has been carefully reviewed, and typographical errors have been corrected.

- Why was R. haemaphysalis not found in cattle as reported in a previous study? Please discuss.

Answer:

Thank you for your observation. The occurrence of R. haemaphysaloides is now addressed in the Discussion section of the manuscript (lines 273–282), including possible reasons for its absence in our cattle samples, supported by relevant references.

- This type of research has previously been reported, particularly focusing on northeastern Thailand. The authors should provide a clearer explanation and strengthen the manuscript by comparing their data with findings from previous studies or by highlighting how their study adds

new insights beyond existing reports in the Discussion section.

Answer:

We aimed to provide comprehensive knowledge in the manuscript by comparing our findings with those of previous publications, particularly Tantrawatpan et al. (2022). Three main points expand the basic information on hard ticks in Thailand, including:

Geographic coverage – While Tantrawatpan et al. (2022) analyzed samples restricted to the northeastern region, this study collected samples from all regions of the country, thereby providing broader and more representative information than data obtained from a limited area.

Expanded molecular analyses – Tantrawatpan et al. (2022) conducted molecular investigations of ticks using nucleotide diversity, haplotype analyses, and phylogenetic analysis. In addition to all the analyses performed by Tantrawatpan et al. (2022), our study further examined the population genetic structure of ticks using Analysis of Molecular Variance (AMOVA), Median-Joining (MJ) network analysis, population pairwise differences (pairwise FST, the genetic exchange parameter (Nm), and demographic history through neutrality indices. These additional analyses, which were not presented in Tantrawatpan et al. (2022), provide deeper insights into the genetic diversity and evolutionary history of tick clades previously reported in Thailand.

Broader interpretation and future implications – Based on the results of these analyses, our study not only presents the current status of tick populations in Thailand but also predicts their historical demographic patterns. Furthermore, we provide a reasoned forecast of potential future trends in tick populations and highlight points of concern, particularly in relation to gene flow dynamics and Thailand’s western trade routes.

In response to the comment, the objectives were clearly summarized and added to the last paragraph of the Introduction section (lines 85–91). The authors have aimed to expand upon the knowledge of ticks in Thailand presented by Tantrawatpan et al. (2022) and other previous publications, as outlined in the manuscript. We believe that the information provided in this study will be valuable for those involved in disease control and livestock production in Thailand, and potentially across Southeast Asia.

Reviewer #3:

The manuscript presents molecular characterization data of Rh. microplus and Hae. bispinosa ticks; however, my main concern, why Rh. sanguineus are given less attention in the study although its huge veterinary importance worldwide. I would say to present the full data

for this specific species as well.

Response to Reviewer #3:

Answer:

Thank you for your valuable observation. This study was not designed to focus on any single tick species; therefore, analyses were conducted on all species for which data were available. In this study, COI sequences were successfully amplified and sequenced from R. microplus (94 sequences) and H. bispinosa (12 sequences); consequently, analyses were performed only on these two species. While previous reports acknowledge the significant impact of R. sanguineus on veterinary health and livestock production, our goals were to identify the Haemaphysalis species and R. microplus clades found across Thailand. Thus, molecular characterization data for the small sample size of R. sanguineus were not obtained in this study and a detailed analysis of this species could not be included in the manuscript (line 165).

The additional requirements have been addressed as described below:

1. The manuscript style was revised to comply with PLOS ONE requirements.

2. The full ethics statement, including the complete name of the ethics committee and the approval number, has been added to the ‘Methods’ section (line 99 – 103) and as shown below.

Ethics Statement: All animal care and experimental

---

## [Decision Letter · Decision Letter 1]

18 Sep 2025

Dear Dr. JITTAPALAPONG,

Thank you for submitting your manuscript to PLOS ONE. After careful consideration, we feel that it has merit but does not fully meet PLOS ONE’s publication criteria as it currently stands. Therefore, we invite you to submit a revised version of the manuscript that addresses the points raised during the review process.

**ACADEMIC EDITOR:** There are minor comments from the reviewer to shape the paper for acceptance.** **

We look forward to receiving your revised manuscript.

Kind regards,

Shawky M Aboelhadid, PhD

Academic Editor

PLOS ONE

Journal Requirements:

Reviewers' comments:

Reviewer's Responses to Questions

**Comments to the Author**

Reviewer #1: (No Response)

Reviewer #3: All comments have been addressed

2. Is the manuscript technically sound, and do the data support the conclusions?

Reviewer #1: Yes

Reviewer #3: Yes

3. Has the statistical analysis been performed appropriately and rigorously?

Reviewer #1: N/A

Reviewer #3: Yes

4. Have the authors made all data underlying the findings in their manuscript fully available?

Reviewer #1: Yes

Reviewer #3: Yes

5. Is the manuscript presented in an intelligible fashion and written in standard English?

Reviewer #1: Yes

Reviewer #3: Yes

Reviewer #1: Thanks for the revision. I believe the Malaysian clade C sequences should be included, as they represent the reference or 'type' sequences.

Reviewer #3: The authors have made the required improvements to their manuscript. It is acceptable for publication now. Good luck!

**Do you want your identity to be public for this peer review?** For information about this choice, including consent withdrawal, please see our Privacy Policy

Reviewer #1: No

Reviewer #3: No

---

## [Author Response · Author response to Decision Letter 2]

3 Nov 2025

We would like to thank you and the reviewers for your time and valuable suggestions. Based on these suggestions, the manuscript has been significantly improved and is now more broadly useful. However, there are additional suggestions regarding certain aspects of the analysis that should be incorporated to further enhance the completeness of the manuscript. The present manuscript has been revised in accordance with the reviewers’ comments, which are addressed in detail below.

Reviewer #1: Thanks for the revision. I believe the Malaysian clade C sequences should be included, as they represent the reference or 'type' sequences.

Answer: We sincerely appreciate your valuable comment. Following this suggestion, the COI nucleotide sequences of R. microplus clade C from Malaysia, available in GenBank (KM246866, KM246867, KM246868, KM246869, KM246870, KM246873, and KM246874), were included in the analysis as reference sequences, along with sequences from Myanmar, Cambodia, Bangladesh, and other countries. These seven new reference sequences were also added to the “Reference Sequences Table” in S1 Appendix. The entire phylogenetic analysis was re-performed using the updated dataset. The newly obtained phylogenetic parameters are presented on page 6 (lines 117–121) and page 11 (lines 180–190) of the manuscript, and the resulting phylogenetic tree is shown in Fig. 1. The topology of the newly obtained phylogenetic tree is generally consistent with the findings of the previous analysis, except for the inclusion of the seven R. microplus clade C sequences from Malaysia within clade C of the tree. Therefore, the new tree fully confirms and supports the molecular identification results reported in the manuscript.

Reviewer #3: The authors have made the required improvements to their manuscript. It is acceptable for publication now. Good luck!

Answer: The authors sincerely thank the reviewer.

Once again, we would like to extend our deep appreciation to the Academic Editor and the reviewers for their thoughtful evaluation, constructive criticism, and encouragement. Their efforts have greatly contributed to enhancing the quality and completeness of our work.

---

## [Editor Report · Decision Letter 2]

5 Nov 2025

Molecular characterization of Rhipicephalus microplus and Haemaphysalis bispinosa ticks from cattle across Thailand: Regional identification and evidence of different genetic sub-structures between mainland and peninsular populations

PONE-D-25-29457R2

Dear Dr.  SATHAPORN - JITTAPALAPONG,

We’re pleased to inform you that your manuscript has been judged scientifically suitable for publication and will be formally accepted for publication once it meets all outstanding technical requirements.

Kind regards,

Shawky M Aboelhadid, PhD

Academic Editor

PLOS ONE
---

## [Editor Report · Acceptance letter]

PONE-D-25-29457R2

PLOS ONE

Dear Dr. Jittapalapong,

I'm pleased to inform you that your manuscript has been deemed suitable for publication in PLOS ONE. Congratulations! Your manuscript is now being handed over to our production team.

Kind regards,

on behalf of

Professor Shawky M Aboelhadid

Academic Editor

PLOS ONE